# Impact of COVID-19 Restrictions on Acute Gastroenteritis in Children: A Regional, Danish, Register-Based Study

**DOI:** 10.3390/children10050816

**Published:** 2023-04-29

**Authors:** Eva Plantener, Kumanan Rune Nanthan, Ulrik Deding, Mads Damkjær, Ea Sofie Marmolin, Lotte Høeg Hansen, Jens J. H. Petersen, Roberto Pinilla, John E. Coia, Donna Lykke Wolff, Zhijun Song, Ming Chen

**Affiliations:** 1Department of Clinical Microbiology, University Hospital of Southern Denmark, 6200 Aabenraa, Denmark; 2Department of Clinical Research, University of Southern Denmark, 5000 Odense, Denmark; 3Department of Surgery, Odense University Hospital, 5700 Svendborg, Denmark; 4Department of Pediatrics, Lillebaelt Hospital, 6000 Kolding, Denmark; 5Institute of Regional Health Research, University of Southern Denmark, 5000 Odense, Denmark; 6Department of Clinical Microbiology, University Hospital of Southern Denmark, 6000 Kolding, Denmark; 7Department of Pediatrics, University Hospital of Southern Denmark, 6200 Aabenraa, Denmark; 8Department of Pediatrics, University Hospital of Southern Denmark, 6700 Esbjerg, Denmark; 9Department of Clinical Microbiology, University Hospital of Southern Denmark, 6700 Esbjerg, Denmark; 10Department of Clinical Research, University Hospital of Southern Denmark, 6200 Aabenraa, Denmark

**Keywords:** gastroenteritis, SARS-CoV-2, children, stool sample, paediatrics

## Abstract

This study aimed to evaluate the impact of the severe acute respiratory syndrome coronavirus 2 (SARS-CoV-2) restrictions such as social distancing on the occurrence of acute gastroenteritis (AGE) among children. This study is a register-based study, including every child seen in the departments of paediatrics with the initial diagnosis of AGE in three neighbouring hospitals in Denmark, from March 2018 through February 2021. The study also included every positive stool sample for AGE-causing pathogens analysed in these three hospitals from children during the same period. The Wilcoxon rank-sum test was used to determine differences between the period during the SARS-CoV-2 restrictions and before. In all, 222,157 children were seen in the three paediatric departments during this period. Of these, 3917 children were diagnosed with AGE. We found a decrease of 46.6% in AGE-related visits per month after the SARS-CoV-2 restrictions were introduced compared to before (*p*-value < 0.001). Positive stool samples decreased by 38.2% (*p*-value = 0.008) during the restrictions. This study found that cases of paediatric AGE decreased significantly the during COVID-19 restrictions, suggesting that studies should be conducted to determine whether this reduction was a result of good hand hygiene and social distancing or just a result of altered health-seeking behaviour among children.

## 1. Introduction

Acute gastroenteritis (AGE) is defined as a rapid onset of diarrhoea and/or vomiting, with or without fever and abdominal pain [1]. The condition is very common, especially in younger children, where the incidence of AGE is estimated at one to five episodes per child per year in children aged under five years. When a child is sick because of AGE, it not only affects the child’s welfare but is also a large burden on the healthcare system and reduces the parents’ ability to work while taking care of the child [2,3]. Most cases of AGE in economically developed countries are benign and self-limiting. The causative microbiological organism is, however, only identified in around 50% of cases of AGE in children, with viruses such as rotavirus being the most common agent in developed countries [4,5].

The transmission of gastrointestinal pathogens is usually faecal–oral, usually through the consumption of contaminated food or water or contact with infected individuals or infected animals, with various transmission routes depending on the causative pathogen [6]. According to the World Health Organization (WHO), key measures such as handwashing with soap, good personal and food hygiene and isolation are important interventions in preventing AGE, and studies have found that improvement in hand hygiene alone can reduce gastrointestinal disease by 30–31% [7,8,9].

On 11 March 2020, the WHO declared a global pandemic due to severe acute respiratory syndrome coronavirus 2 (SARS-CoV-2) [10]. As a response to the emerging pandemic, numerous countries throughout the world implemented various national infection control and prevention measures. These were based on the already established recommendations by the WHO, such as social distancing; closure of schools/day-care centres; lockdown of crowded places; limiting the number of people at gatherings and focus on improved hygiene measures, e.g., hand washing and sanitizing [11]. The implemented restrictions was to reduce the transmission and spread of SARS-CoV-2, but recent studies have also observed a decrease in the incidences of several other infectious diseases during the timeframe of the SARS-CoV-2 pandemic. Studies have reported a reduction of 63% in acute otitis media in children in the Netherlands and a reduced risk ratio of 30–87% for salmonellosis, shigellosis and campylobacteriosis in Israel [12,13,14,15].

As a part of the post-pandemic evaluation of the overall cost-effectiveness of these various national infection-control strategies, it is pivotal to examine different aspects of these lockdown strategies. Reporting the incidence of infectious diseases other than SARS-CoV-2 in various countries with different approaches to the pandemic response could help compare which strategy, if any, was most effective against infectious diseases, such as, AGE [16,17,18]. For example, two studies from countries with different strategies, England and Norway, found a 52% and 74% decrease, respectively, in hospital visits due to AGE in children during the SARS-CoV-2 restrictions compared to pre-pandemic times [19,20].

While this study only briefly touches upon the different strategies implemented by each country, it will report the incidence of AGE in children before and after the implementation of SARS-CoV-2 restrictions in Denmark, as this has not yet been reported. Furthermore, this study will examine the incidence of AGE-causing microorganisms in children in the same period. Other studies have reported solely on the incidence of clinical AGE but have not examined whether a reduction in the causative microorganisms responsible for AGE also occurred [21,22,23].

The aim of this study was to examine the effect of SARS-CoV-2 restrictions on the incidence of AGE in children in three hospitals in the Region of Southern Denmark, by comparing registered hospital visits to the paediatric department caused by acute gastroenteritis and the incidence of positive stool samples before and during the restrictions.

## 2. Materials and Methods

### 2.1. Approvals

This study was approved by the Region of Southern Denmark (journal number 21/32259). Since this is a retrospective, register-based study with thousands of children involved, we were authorized to include the participating children without asking for individual consent. The study was also approved by the heads of the participating departments and hospitals.

### 2.2. Outcome

The primary outcome of this study was to compare the incidence of visits to the department of paediatrics caused by AGE before and during the SARS-CoV-2 restrictions in three regional hospitals in Southern Denmark. The secondary outcome of the study was to assess the number of stool samples positive for AGE-causing pathogens in children before and after the introduction of SARS-CoV-2 restrictions.

### 2.3. Study Period

The study included children from 1 March 2018 through 28 February 2021. SARS-CoV-2 restrictions were implemented on 11 March 2020 in Denmark, and most of the restrictions were lifted on 28 February 2021. Please refer to Figure 1 for a timeline of the restrictions in Denmark. We defined three time periods in this study, each consisting of 12 months: control period 1 (March 2018 through February 2019), control period 2 (March 2019 through February 2020) and the period during SARS-CoV-2 restrictions (March 2020 through February 2021). For practical reasons, the first 10 days of March were included in the restriction period though the restrictions were not yet implemented. 

### 2.4. Population and Data

This study was conducted as a multicentre, retrospective, register-based study, including every child (aged 0–17) seen during the study period at the department of paediatrics due to AGE in three hospitals, Lillebaelt Hospital (SL), Hospital of Southern Jutland (SHS) and Hospital South West Jutland (SVS), in the Region of Southern Denmark as well as all positive stool samples from children analysed in the department of microbiology in the same hospitals during the same time period.

The catchment area of these three hospitals combined includes 760,000 citizens, corresponding to approximately 15% of the population of Denmark [24]. Southern Denmark has been shown to be representative of most socioeconomic and healthcare aspects, when compared to the entire Danish population [25].

In Denmark, all children must be referred to the department of paediatrics by a general practitioner or via the emergency number in each region, except in case of acute illness, where an ambulance is needed [26]. The department of paediatrics in each of the three hospitals provided a list of every visit to their department from 1st March 2018 through 28 February 2021, obtained through the electronic patient journals. This list also included the date of the visit, social security number, sex and age of the children as well as the initial diagnosis received in the department of paediatrics.

Children seen at the department of paediatrics receive an initial diagnosis whether they are seen in the outpatient clinic, sent home from the emergency paediatric department, or admitted. This study included the AGE-related diagnoses according to the International Classification of Diseases (ICD-10) A08 (viral and other specified intestinal infections), A09 (other gastroenteritis and colitis of infectious and unspecified origin), E86 (volume depletion) and R11 (nausea and vomiting) [27].

All stool samples analysed in Denmark are reported in the Department of Microbiology’s Data Systems (MADS) database [28]. The departments of microbiology in the three hospitals obtained every positive stool sample from this database for children aged 0–17 from the study period. The departments of microbiology at the hospitals also analyse stool samples from the primary care physician in the catchment area as well as stool samples from the hospital itself. In Denmark, it is not recommended to analyse stool samples regularly in case of diarrhoea, unless one or more of the following characteristics are present: severe, long-lasting (>14 days), bloody or travel-associated diarrhoea; diarrhoea in new-borns or suspicion of haemolytic uraemic syndrome (HUS) [29]. It can also be beneficial in cases where the physician suspects an outbreak [30]. As mentioned in the introduction, most cases of AGE are caused by viral agents that are self-limiting, and a stool sample is thus not indicated in most cases [5].

A positive stool sample was defined in this study as having strains of *E. coli*, *C. difficile*, *Shigella* spp., *Campylobacter* spp., *Salmonella* spp., *Yersinia* spp., norovirus, astrovirus, sapovirus, rotavirus, adenovirus, *Giardia* spp., *Cryptosporidium* spp. and *Entamoeba histolytica*. In case of a sample that was positive for more than one bacterium, virus or parasite, the sample was only counted once.

Unfortunately, the total number of analysed stool samples was not available, and therefore, rates of positive stool samples are not reported in this study.

The original data might be obtained by contacting the authors.

### 2.5. Statistical Analysis

The total number of visits to the departments of paediatrics and the number of positive stool samples were reported monthly by the hospitals, and visits with the initial diagnosis of AGE were presented as a proportion of total visits due to all causes.

The average number of visits to the department of paediatrics and the average number of positive stool samples before and during the SARS-CoV-2 restrictions were calculated. Children were divided into two age groups (0–5 years and >5 years) as the prevalence and severeness of AGE differs between younger and older children [31,32]. Chi-squared tests were used to determine statistical significance in baseline characteristics. Wilcoxon rank-sum tests were conducted comparing hospital admission and positive stool samples in control period 2 and the restriction period with control period 1 as a baseline [33]. A *p*-value less than 0.05 was considered significant. All statistical analyses were performed with STATA (version 15.0, StataCorp LLC, College Station, TX, USA) [34,35].

## 3. Results

### 3.1. Primary Outcome

From 1 March 2018 through 28 February 2021, 222,157 children were seen in the departments of paediatrics in Lillebaelt Hospital (SL), Hospital of Southern Jutland (SHS) and Hospital South West Jutland (SVS). Please refer to Table 1 for baseline characteristics. In total, 78,515 visits were registered in control period 1, 71,193 visits in control period 2 and 72,449 in the restriction period. Of these, 3917 children (1.8%) received the initial diagnosis of AGE divided between 1476 (1.9%) in control period 1, 1615 (2.3%) in control period 2 and 826 (1.1%) in the restriction period. The number of AGE-related visits peaked with 238 visits in April 2019 during control period 2, and the lowest numbers of AGE-related visits were reported in April 2020 during the restriction period (45 visits). Please refer to Figure 1 for monthly visits. There was an equal distribution between boys (51.6%) and girls (48.4%) both during the control periods and the restriction period (*p* = 0.901). 

The majority of the patients seen due to AGE pre-pandemic were aged between 0 and 5 years (78.8%), whilst this proportion decreased to 68.5% during the restrictions (*p* < 0.001).

The monthly average of the initial diagnoses of AGE was 2.1% of all visits to the department of paediatrics during the control periods and 1.1% during the restriction period (Table 2 and Figure 2). These changes were significant (*p* < 0.001).

The monthly number of initial diagnoses of AGE varied from 89 to 181 during the control periods, while it was 45–106 during the restriction period, with March 2022 being the month with the highest number during these 12 months (Figure 2).

In the control periods, there was an average of 128.8 initial diagnoses of AGE per month; this number decreased to 68.8 during the period with SARS-CoV-2 restrictions (Table 2 and Figure 2), corresponding to a decrease of 46.6%. The Wilcoxon rank-sum test indicated no significant differences in median visits to the paediatric department between control period 1 and 2, whereas the period with SARS-CoV-2 restrictions differed significantly from control period 1 (Table 2, *p* < 0.001).

### 3.2. Secondary Outcome

From 1 March 2018 through 28 February 2021, the departments of microbiology at Lillebaelt Hospital (SL), Hospital of Southern Jutland (SHS) and Hospital South West Jutland (SVS) reported 1261 positive stool samples for AGE-causing pathogens in children, 413 positive stool samples in control period 1, 550 positive stool samples in control period 2 and 298 positive stool samples in the restriction period.

The proportion of samples from boys and girls and children aged > 5 years did not differ significantly between the control periods and the restrictions period (Table 1, *p* = 0.008 for both). The characteristics of the children with positive stool samples can be seen in Table 1.

The number of positive stool samples per month varied from a low of 16 in May 2019 to a peak of 110 in August 2019 during the control periods, whilst the number of monthly positive stool samples in the restriction period varied from a low of 12 in April 2020 to a peak of 44 in October 2020 (Figure 3).

During the control periods, there was an average of 40.1 positive stool samples per month, while this number decreased to 24.8 per month during the restriction period (Table 2 and Figure 3), corresponding to a decrease of 38.2%. The Wilcoxon rank-sum test found a significant difference in the median number of positive stool samples between control period 1 and 2, as well as a significant difference during the restrictions period compared to control period 1 (Table 2, Figure 3).

## 4. Discussion

This study found a 46.4% decrease in AGE-related visits to the departments of paediatrics in three hospitals in the Region of Southern Denmark after the introduction of SARS-CoV-2 restrictions as compared to the two years prior. This decrease was statistically significant (*p* < 0.001). The decline agrees with similar studies that found a decrease in visits due to AGE of 52–73% during the pandemic [21,23,36].

The percentage of AGE-related visits to the department of paediatrics also showed a statistically significant decline after SARS-CoV-2 restrictions were introduced (2.1% during the control period vs. 1.1% during the SARS-CoV-2 restrictions), suggesting that the decline in AGE-related visits cannot be explained by fewer visits to the department of paediatrics in general.

This study also found a significant 38.2% decrease in children of stool samples that were positive for AGE-causing pathogens per month during the SARS-CoV-2 restrictions as compared to before. Studies examining the prevalence of stool samples positive for AGE-causing pathogens have found similar results. U Eigner et al. found a significant decrease in norovirus in 2020 among hospitalized patients in Germany. These data, however, only concern norovirus and include adults [37]. Bassal et al. also found a reduced risk ratio of 30–87% for Campylobacter, Salmonella, and Shigella in all age groups in Israel during the SARS-CoV-2 pandemic [15]. Likewise, a study similar to ours was recently conducted in England by N K Love et al., which reported a decrease in reported GI infections by 52% between February and July 2020 and a decrease of 34% in laboratory-confirmed cases; however, the study only included the first six months of the pandemic and only included adults [20]. Finally, a study from Oslo University Hospital in Norway by Knudsen et al. compared GI cases between pre-pandemic years and the SARS-CoV-2 pandemic period between 1 April 2020 and 31 March 2021 and found a decrease of 74% in gastroenteritis diagnoses during the pandemic period and found significantly lower levels of stool samples positive for adeno- and norovirus during the pandemic period compared to previous levels [19].

An interesting facet of the studies is that the national SARS-CoV-2 restriction strategies differed between the countries. England gradually introduced SARS-CoV-2 restrictions on 16 March 2020 by introducing social distancing and implementing further restrictions such as school closures and work from home. On the 23rd of March, a partial lockdown was implemented UK-wide. On 4 July 2020, pubs and restaurants reopened, and there was a gradual easing of restrictions [20]. In contrast, Norwegian restrictions were similar to those implemented by the Danish government by implementing a six-week nationwide lockdown from 12 March 2020 and maintaining infection-control measures throughout most of the pandemic until late September 2021 [19]. Though the strategies varied, both studies found a significant decrease in AGE contacts and GI-associated faecal tests, with the Norwegian study finding a greater decrease. Despite differences in restriction severity and timing, the focus on good hand hygiene and social distancing seemed to be effective.

Visits caused by AGE to the department of paediatrics in our study showed a seasonal pattern, with visits increasing in winter and early spring (Figure 2). This tendency seemed to flatten out during the SARS-CoV-2 restrictions. A similar decrease in hospital visits has been shown for other health conditions after the introduction of SARS-CoV-2 restrictions [38,39]. While the data largely follow a trend of seasonal patterns (Figure 2 and Figure 3), in the spring and summer of 2019, we found spikes in hospital visits due to AGE and positive stool samples without any obvious cause. These spikes could have led to an underestimation of the decrease in positive stool samples in the statistical analysis.

This study had a large sample size due to using well-documented electronic databases; however, the control period available for comparison with the restrictions period was limited to two years in this study. While the Wilcoxon rank-sum test did not find a statically significant difference in visits when comparing control period 1 and 2, it did find a difference in positive stool samples when comparing the two control periods, and the periods could differ from previous years in terms of AGE incidence, leading to over- or underestimation of AGE during the SARS-CoV-2 restrictions. The inclusion of the first 10 days of March 2020 in the restriction period could also have impacted the results, as the restrictions were not in effect yet. This could underestimate our effect size given our hypothesis that the incidence of AGE decreased when the restrictions were implemented. In Denmark, Statens Serum Institut, a governmental public health and research institution under the Danish Ministry of Health, monitors epidemics across the country and investigates cases of spikes in infectious diseases. While there were a few outbreaks of AGE-causing agents in 2019, very few cases were in the Region of Southern Denmark, and even less were among children (less than 10 cases in 2019), making this unlikely to have been a factor in our study [40].

Another weakness of this study is the unknown number of children with AGE who did not seek medical treatment during the SARS-CoV-2 restrictions, as this number might have been higher during the restrictions. It is possible that parents were more likely to keep their children with mild-to-moderate AGE symptoms at home during the pandemic due to the fear of contracting SARS-CoV-2 during their visit to the hospital [41,42]. Patient behaviour and healthcare prioritization during the SARS-CoV-2 pandemic may also have affected the number of referrals to the departments of paediatrics due to AGE and the number of stool samples sent for analysis. Especially during the first national lockdown, studies have indicated lower admission rates in the emergency departments [43,44]. Our study did not analyse behaviour during the restrictions, and as the total number of stool samples analysed was not available in this study, the decrease in positive stool samples could possibly be caused by fewer analysed stool samples in general. Thus, this study cannot conclude that hand washing and social distancing alone prevent visits to the paediatric departments and curtail the spread of AGE-causing pathogens. Limited travel, changes in referrals to the hospitals and hesitance to go to said hospitals are just a few examples of how society was different during the year of SARS-CoV-2 restrictions [41,42,43,44,45].

This study does, however, indicate that infection-control measures such as hand washing and social distancing may contribute to a reduction in visits to the department of paediatrics with symptoms of gastroenteritis as well as in AGE-causing pathogens. Therefore, in the case of outbreaks of gastroenteritis, a childcare institution can enforce the guidelines that were implemented during the SARS-CoV-2 pandemic to prevent further disease [46].

Future studies should include more pre-pandemic years as well as the post-pandemic period to examine how the incidence of AGE changed when the restrictions were lifted. Furthermore, to test the effectiveness of hand washing and social distancing, these future studies should also examine hand washing and social distancing compliance during restrictions.

## Figures and Tables

**Figure 1 children-10-00816-f001:**
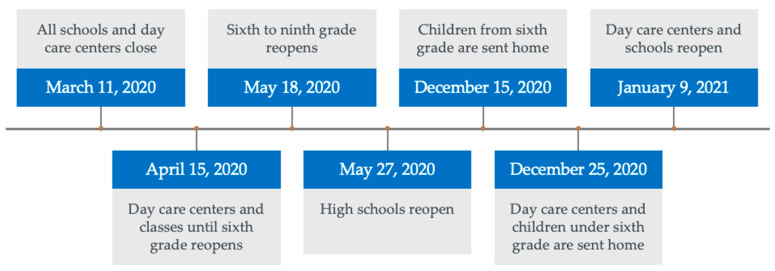
SARS-CoV-2 restrictions in Denmark with a focus on children.

**Figure 2 children-10-00816-f002:**
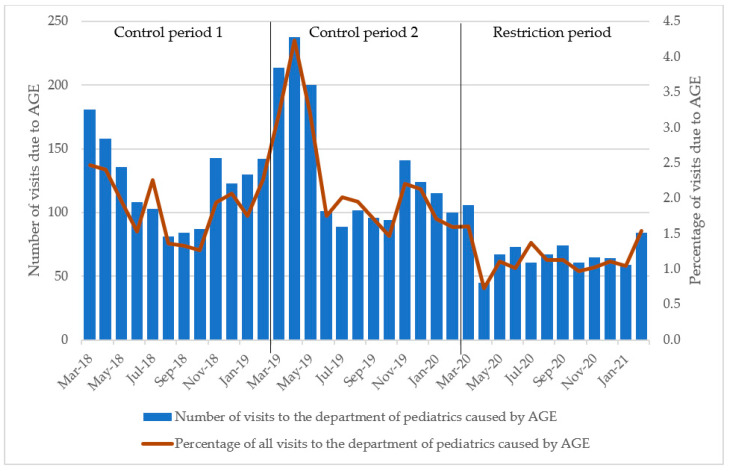
Number of visits to the department of paediatrics due to acute gastroenteritis in children. Percentage of all visits due to AGE during the same time. Control period 1 refers to March 2018 through February 2019, control period 2 refers to March 2019 through February 2020 and the restriction period refers to March 2020 through February 2021. Abbreviations: AGE, acute gastroenteritis.

**Figure 3 children-10-00816-f003:**
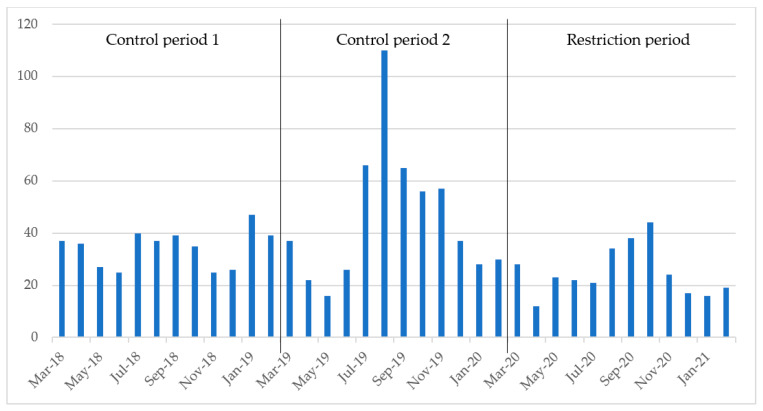
Number of positive stool samples per month. Control period 1 refers to March 2018 through February 2019, control period 2 refers to March 2019 through February 2020 and the restriction period refers to March 2020 through February 2021.

**Table 1 children-10-00816-t001:** Baseline characteristics divided between visits to the department of paediatrics and stool samples positive for pathogens causing acute gastroenteritis in children. Control period 1 refers to March 2018 through February 2019, control period 2 refers to March 2019 through February 2020, the restriction period refers to March 2020 through February 2021 (SARS-CoV-2 restrictions). * denotes children without a Danish security number. Abbreviations: AGE, acute gastroenteritis.

	Visits to the Department of Paediatrics Due to AGE (*n* = 3917)				Stool Samples Positive for Pathogens Causing Acute Gastroenteritis (*n* = 1261)			
	Control Period 1 (*n* = 1476)	Control Period 2 (*n* = 1615)	Restriction Period (*n* = 826)	*p*-Value	Control Period 1 (*n* = 413)	Control Period 2 (*n* = 550)	Restriction Period (*n* = 298)	*p*-Value
**Sex**								
Male	755 (51.2)	838 (51.9)	429 (51.9)		211 (51.1)	290 (52.7)	180 (60.4)	
Female	721 (48.2)	777 (48.1)	397 (48.1)	0.901	199 (48.2)	260 (47.3)	108 (36.2)	0.008
*Missing*	0 (0.0)	0 (0.0)	0 (0.0)		3 * (0.7)	0 (0.0)	10 * (3.4)	
**Age**								
0–5 years	1166 (79.0)	1268 (78.5)	567 (68.6)		253 (61.3)	389 (70.7)	202 (67.8)	
>5 years	310 (21.0)	347 (21.5)	259 (31.4)	<0.001	160 (38.7)	161 (29.3)	96 (32.2)	0.008

**Table 2 children-10-00816-t002:** Changes in medians. Visits to the paediatric departments and positive stool samples in the study period. Wilcoxon rank-sum tests comparing monthly admissions for AGE and positive stool samples from control period 2 (March 2019 through February 2020) and the restriction period (March 2020 through February 2021) with the baseline being control period 1 (March 2018 through February 2019) for both.

	Control Period 1	Control Period 2	*p*-Value	Restriction Period	*p*-Value
Visits to the department of paediatrics, total	78,515	71,193		72,449	
Visits due to AGE, total (%)	1476 (1.9)	1614 (2.3)		826 (1.1)	
Positive stool samples for AGE-causing pathogens, total	413	550		298	
Visits due to AGE, monthly median (IQR range)	126.5 (81–181)	108.5 (89–238)	0.977	66 (45–106)	<0.001
Positive stool samples for AGE-causing pathogens, monthly median (IQR range)	36.5 (25–47)	37 (16–110)	0.485	22.5 (12–44)	0.008

## Data Availability

The data presented in this study are available on request from the corresponding author. The data are not publicly available due to GDPR policies.

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
