# Peer review of "Impact of COVID-19 Restrictions on Acute Gastroenteritis in Children: A Regional, Danish, Register-Based Study"

_children, 2023, doi:10.3390/children10050816_

Round 1

Reviewer 1 Report

This is a multi-centre retrospective register-based study, including three hospitals in the Region of Southern Denmark about Impact of COVID-19 restrictions on acute gastroenteritis in children.
The structure of the article is correct and the data is presented clearly.
Figure 1 must be moved under the text in which it is explained.
The article presents 24 references, being up to date. I recommend you to expand the discussions, so you will have a larger number of references. Please arrange reference 15. Please complete the year when the links from references 13 and 16 last accessed.

Author Response

Dear reviewer,

Thank you for your comments and questions.

Point 1: Figure 1 must be moved under the text in which it is explained.

Answer: Thank you for pointing the out. We have made this change.

Point 2: The article presents 24 references, being up to date. I recommend you to expand the discussions, so you will have a larger number of references.

Answer: Thank you for this insight. We have expanded by 20 references.

Point 3: Please arrange reference 15.

Thank you for pointing this out. We have made this change.

Point 4: Please complete the year when the links from references 13 and 16 last accessed.

Thank you for pointing this out. We have made this change to the manuscript.

Reviewer 2 Report

The authors conducted a register based study to examine the effect of SARS-CoV-2 restrictions on the incidence of AGE in children from Southern Denmark. The major limitations of the study have already been pointed out by the authors. The decrease in AGE related visits may simply be related to the decrease in children/parents seeking care during the pandemic and may not reflect an actual decrease in the incidence of AGE. The decrease in the positivity rate of stool samples may simply reflect a biased population who still elect to seek care despite pandemic restrictions, which is certainly a confounding factor. AGE related visits or stool sample positivity are therefore not good surrogates for AGE incidence. With these significant limitations, it is impossible to draw conclusions on the effects of pandemic restrictions on AGE incidence, and the conclusion of this paper made by the authors that pandemic restrictions (hand hygiene/social distancing) can contribute to prevention of AGE is not supported given the significant limitations.

Author Response

Dear reviewer,

Thank you for your comments.

Point 1: With these significant limitations, it is impossible to draw conclusions on the effects of pandemic restrictions on AGE incidence, and the conclusion of this paper made by the authors that pandemic restrictions (hand hygiene/social distancing) can contribute to prevention of AGE is not supported given the significant limitations.

Answer: Thank you for pointing this important point out. We have discussed this comment in our group and agree that the conclusion might have been too confident. We have changed the conclusion to: “This study found a significant decrease in hospital visits due to acute gastroenteritis in children during the SARS-CoV-2 restrictions in three hospitals in Denmark as well as in stool samples positive for AGE-causing pathogens. While society changed in numerous ways during the pandemic, this study suggests that good hand hygiene and social distancing contributes to prevention of acute gastroenteritis, helping to prevent the spread of this disease in childcare institutions in the future” (lines 286-291).

Reviewer 3 Report

* The conclusion is not justified by the methods and the results.

* Tables and figures: Quality of figures is so important too. Please provide some high-resolution figures. Some figures have a poor resolution.

Author Response

Dear reviewer,

Thank you for your comments.

Point 1: The conclusion is not justified by the methods and the results.

Answer: Thank you for pointing this important point out. We have discussed this comment in our group and agree that the conclusion might have been too confident. We have changed the conclusion to: “This study found a significant decrease in hospital visits due to acute gastroenteritis in children during the SARS-CoV-2 restrictions in three hospitals in Denmark as well as in stool samples positive for AGE-causing pathogens. While society changed in numerous ways during the pandemic, this study suggests that good hand hygiene and social distancing contributes to prevention of acute gastroenteritis, helping to prevent the spread of this disease in childcare institutions in the future” (lines 286-291).

Point 2: Tables and figures: Quality of figures is so important too. Please provide some high-resolution figures. Some figures have a poor resolution.

Answer: Thank you for pointing this out. We have made changes to the figures and hope this solved the problem.

Reviewer 4 Report

This is a retrospective study of children’s visits to the department of paediatrics of three hospitals in the Region of Southern Denmark between March 2018 and February 2021 to determine the number of visits with respect to acute gastroenteritis and the number of stool samples positive for pathogens causing acute gastroenteritis. The results indicated that, during the period of restrictions as a result of COVID-19, both numbers went down. The authors conclude that hand washing and social distancing contributed to the reduction and should be encouraged to maintain this result.

The strengths of this paper are that it is well written, asking the right type of questions and devising the appropriate study to determine the sought after result. The weaknesses relate to the initial information provided in the Introduction regarding acute gastroenteritis not only being out of date but not taking into consideration COVID-19. As well, there are a number of claims that are not referenced. Figure 1 needs to be reformatted to be clear. Other concerns are mentioned in the line by line suggested edits to follow.

39-47 In a paper concerned with the role of measures undertaken as a result of COVID-19 that may have helped reduce the incidence of AGE, it is important to cite references regarding AGE that have been published since 2019. Rather than citation 1, a reference from 2007 on AGE, this is an updated reference in this regard:

Guarino, A.; Aguilar, J.; Berkley, J.; Broekaert, I.; Vazquez-Frias, R.; Holtz, L.; Lo Vecchio, A.; Meskini, T.; Moore, S.; Rivera Medina, J.F.; Sandhu, B.; Smarrazzo, A.; Szajewska, H.; Treepongkaruna, S. (2020). Acute Gastroenteritis in Children of the World: What Needs to Be Done? J. Pedia. Gastroenter. Nutri. 202070, 694–701. https://doi.org/10.1097/MPG.0000000000002669.

Information from 2007 cannot be used regarding the incidence of diarrhea in Europe during COVID-19. Here is a reference that is appropriate that the authors can consider:

Wang, J.; Yuan, X. Digestive system symptoms and function in children with COVID-19: A meta-analysis. Med.  2021,100, e24897. https://doi.org/10.1097/MD.0000000000024897.

Similarly, with respect to the burden of AGE on the healthcare system and on parents, a current reference that the authors may substitute is:

Cohen, R.; Martinón-Torres, F.; Posiuniene, I.; Benninghoff, B.; Oh, K.-B.;  Poelaert, D.The Value of Rotavirus Vaccination in Europe: A Call for Action. Infect. Dis. Ther. 202312, 9–29. https://doi.org/10.1007/s40121-022-00697-7 

Information from the WHO cannot be from 2008, as are both references 4 and 5. Please update with current information from the WHO.

52-56 Need references regarding the various responses to COVID-19 of countries with respect to infection control and prevention. 

63-64 “each country chose a different approach to the pandemic”—need references specifically for the different responses chosen by England, Norway and Denmark.

64-65 “other studies have reported”—please reference these studies.

83 Change “figure 1” to “Figure 1”.

88 Figure 1 should follow this paragraph, since this is where it is mentioned.

91-93 Need a reference for this claim that most patients in Denmark are assessed by primary care physicians  before referral to the department of paediatrics.

106-107 “All stool samples analysed in Denmark are reported in the Department of Microbiology’s Data Systems (MADS) database.”—need a reference for this claim.

114-115 “most cases of AGE are caused by viral agents that are self-limiting”—need a reference for this claim.

121-122 Please increase the font size of the information provided above each of the dates. It is much too small in comparison to the dates and cannot be easily read. As well, increase the size of each of the rectangles so that the dates of each can fit on one line. The font size of the dates can be reduced as well. The dates do not need to be so large.

126 Do the authors mean that the hospitals reported the total number of visits monthly or that the researchers engaged in this study did so? Please state who did the reporting.

130-131 Need a reference for the pathogens differing between younger and older children.

135-136 Please explain why STATA (version 15.0) was used for the statistical analyses and provide a reference to indicate that it has been used as such for similar recent research.

143 and 147 Table 2 is mentioned before Table 1. Either rename Table 2 as Table 1 and vice versa or else mention the current Table 1 before Table 2 is mentioned.

154-155 Please decrease the spaces between each of the columns so that “Restriction” can appear on one line and not force the “n” onto the second line.

156-160 Title for Table 1 should appear before Table 1, not after.

156-157 Change “Baseline characteristics. Column one concerns visits to the department of paediatrics. Column two concerns stool samples positive for pathogens causing acute gastroenteritis in children” to “ Baseline characteristics divided between visits to the department of paediatrics and stool samples positive for pathogens causing acute gastroenteritis in children”

163-166 Title for Table 2 should appear before Table 2, not after.

198-200 Were there comparative studies regarding stool samples, as there were for AGE-related visits to the department of paediatrics? Please check the following Google Scholar search and provide the references, if there were: https://scholar.google.ca/scholar?hl=en&as_sdt=0%2C5&q=decrease+of+stool+samples+positive+for+acute+gastroenteritis-causing+pathogens+during+COVID-19&btnG.

226 Change “figure 2” to “Figure 2”.

264-267 Each of these ways mentioned that society changed needs a reference.

268-271 Given that the authors say that both hand washing and social distancing can contribute to prevention, the authors cannot just recommend hand washing in children here without stating why they are not recommending social distancing. Furthermore, in not recommending social distancing, the authors have to provide references for why this is not recommended.

272-273 “the institution can enforce the guidelines that were implemented during the SARS-CoV-2-restrictions pandemic to prevent further disease”—which institution? Which guidelines” Please provide a reference for these guidelines.

327 Please include the year the article was accessed.

332 Please remove “StatPearls Publishing Copyright © 2022” from the reference.

334 Please include the year the article was accessed.

350 Please provide the full journal reference and include the year the article was accessed.

Reviewer 5 Report

Dear authors,

I have completed the review of the manuscript titled “Impact of COVID-19 restrictions on acute gastroenteritis in children: A regional Danish register-based study.”

In the present study, the authors evaluated the impact of the severe acute respiratory syndrome coronavirus (SARS-CoV-2) restrictions such as social distancing on the occurrence of acute gastroenteritis (AGE) among children.

The manuscript is interesting and, in general, well-written.

I have some suggestions to further improve the quality of the manuscript:

1. The Introduction section introduced some relevant articles. Please explain the results or summarize with effect sizes.

2. I suggest that the authors clarify how other researchers can obtain the original data.

3. The authors used diagnosis codes like DA08. It seems like it is neither ICD-10 nor ICD-9. Could you explain what these coding systems are?

4. In the ‘Statistical analysis’ section, it would be better to refer to one or two articles about statistical methods selection to justify your methods. For example, statistical standard about methods for testing statistical differences between groups in medical research, or regression analysis for continuous independent variables will be proper fit to this articles' methods.

5. What is the future scope of the proposed research? The authors have described the limitations in a good way, and I suggest that these can be the future scope of the work.

Author Response

Dear reviewer,

Thank you for your comments.

Point 1: The Introduction section introduced some relevant articles. Please explain the results or summarize with effect sizes.

Answer: Thank you for this comment. We have added the following: Other than reducing SARS-CoV-2 transmission, studies have shown a decrease in the incidences of other infectious diseases during the period of SARS-CoV-2 pandemic, i.e., a reduction in acute otitis media of 63% and a reduced risk ratio of 30-87% of confirmed gastroenteritis [11-14] (lines 54-57).

Point 2: I suggest that the authors clarify how other researchers can obtain the original data.

Answer: Thank you for this suggestion. We have added the following sentence: “Original data can be obtained by contacting the authors.” (line 127).

Point 3: The authors used diagnosis codes like DA08. It seems like it is neither ICD-10 nor ICD-9. Could you explain what these coding systems are?

Answer: Thank you for pointing this out! We have indeed used ICD-10 codes, however we understand how this could be confusing. We have updated the part to: “…this study included the AGE-related ICD-10 diagnoses A08 (viral and other specified intestinal infections), A09 (other gastroenteritis and colitis of infectious and unspecified origin), E86 (volume depletion), R11 (nausea and vomiting)” (lines 103-106).

Point 4: In the ‘Statistical analysis’ section, it would be better to refer to one or two articles about statistical methods selection to justify your methods. For example, statistical standard about methods for testing statistical differences between groups in medical research, or regression analysis for continuous independent variables will be proper fit to this articles' methods.

Answer: Thank you for your comment. We have added a reference to line 138.

Reference: Mishra, P.; Pandey, C.M.; Singh, U.; Keshri, A.; Sabaretnam, M. Selection of appropriate statistical methods for data analysis. Ann. Card. Anaesth. 2019, 22, 297-301, doi:10.4103/aca.ACA_248_18.

Point 5: What is the future scope of the proposed research? The authors have described the limitations in a good way, and I suggest that these can be the future scope of the work.

Answer: We added the following to the discussion: “Future studies should include more pre-pandemic years as well as post-pandemic to examine how the incidence of AGE changed when the restrictions were lifted.” (lines 284-285).

Round 2

Reviewer 2 Report

The revised wording still implies a causal relationship between hand hygiene/social distancing and decreased incidence of AGE, which cannot possibly be demonstrated by such a retrospective study.

Simply changing the wording also did not address the issue I mentioned in my previous comment, which is "The decrease in AGE related visits may simply be related to the decrease in children/parents seeking care during the pandemic and may not reflect an actual decrease in the incidence of AGE. The decrease in the positivity rate of stool samples may simply reflect a biased population who still elect to seek care despite pandemic restrictions, which is certainly a confounding factor. AGE related visits or stool sample positivity are therefore not good surrogates for AGE incidence." This significantly undermines the validity of the study results and it is hard to draw any definitive conclusions based on this data. Therefore I must reject this paper. 

Author Response

Dear reviewer,

We understand your viewpoint. We have decided to remove the discussion based on your comments as well as another reviewer. While we understand that this might now change your opinion on rejecting the paper, we believe the article is better for it.

Reviewer 4 Report

This version of the paper regarding AGE in children with respect to three time periods (including one during COVID-19 restrictions) at three hospitals in the Region of Southern Denmark is much improved over the first. However, there remain some minor concerns as well as one substantial concern. The substantial concern is that the authors still conclude that hand washing and social distancing should be encouraged although the study they conducted neither tested for the effects of hand washing nor social distancing on AGE and, therefore, can make no recommendations regarding either. Consequently, the Conclusion, which is not required by the journal in any case, should be eliminated. Further suggested edits are as follows.

34-35  Change “This study suggests that good hand hygiene and social distancing can contribute to prevention of acute gastroenteritis and could prevent the spread of AGE in institutions in the future.” to “This study found that AGE in the institutions decreased significantly during COVID-19 restrictions suggesting that studies should be done to determine if this reduction was a result of good hand hygiene and social distancing.”

36 As this study did not include testing for good hand hygiene or social distancing, “Hygiene” should be eliminated as a Keyword. Furthermore, “Microbiology” is not mentioned in the Abstract and all Keywords must be included in the Abstract. Therefore, change, “Hygiene; Microbiology” to “Stool sample; Paediatrics”.

51-53 Given that this study neither tested for hand washing nor isolation in the children who visited the paediatrics departments in the three hospitals this information is irrelevant and should be eliminated.

69-70 “other studies have reported isolated on incidence of clinical AGE”—this is not correct English. Please reword.

87 Mention should be made that the study began ten days before the COVID-19 restrictions came into effect. That the study began before the lockdown will need to be recognized later in the limitations section as a limitation of the study.

92 Change “where the restrictions” to “when the restrictions”.

93 Change “where most restrictions” to “when most restrictions”.

95-96 Figure 1 is much improved. Please center justify the information in the grey boxes.

114 What is an ICD-10 diagnosis? Please explain.

182-183 Although this is not stated directly on the Word template for Children, entries in tables for all MDPI journals are expected to be center justified, not left justified. Please reformat Table 1 so that all columns are center justified.

196-197 Please reformat Table 2 so that all columns are center justified.

241 Change “concers” to “concerns”.

252 Change “has only” to “only”

253 Change “has also” to “also”.

313-314 As the authors have noted in their limitations, it is possible that children may have been kept at home rather than taken to the hospital during COVID-19 if they were experiencing AGE. As a result, especially since the authors did not test whether the children seen had actually increased their hand washing and employed social distancing, the authors cannot conclude that hand washing and social distancing can contribute to the prevention of hospital visits regarding AGE. What the authors can say is that they “may” contribute.

315-317 Change “Hand washing and distancing among children should be encouraged in child caring institutions and in case of outbreaks of gastroenteritis, the institution can enforce the guidelines” to “Therefore, in the case of outbreaks of gastroenteritis, the institution can enforce the guidelines”.

320 Change “restrictions were lifted.” to “restrictions were lifted. Furthermore, to test the effectiveness of hand washing and social distancing, these future studies should also examine hand washing and social distancing during restrictions."

Please delete the Conclusion. Based on the data gathered, no conclusion can be drawn about hand washing nor social distancing.

359-360  There is no journal nor date mentioned for reference 5. Please include this information.

386-391 There is no journal nor date mentioned for references 17 or 18. Please include this information.

403-404 There is no journal nor date mentioned for reference 23. Please include this information. Furthermore, the link provided goes to a 404 page, not the article. Please provide the correct link.

412-413 There is no journal nor date mentioned for reference 27. Please include this information.
